# ON THE ROBUSTNESS OF SENTIMENT ANALYSIS FOR STOCK PRICE FORECASTING

## ABSTRACT

Machine learning (ML) models are known to be vulnerable to attacks both at training and test time. Despite the extensive literature on adversarial ML, prior efforts focus primarily on applications of computer vision to object recognition or sentiment analysis to movie reviews. In these settings, the incentives for adversaries to manipulate the model's prediction are often unclear and attacks require extensive control of direct inputs to the model. This makes it difficult to evaluate how severe the impact of vulnerabilities exposed is on systems deploying ML with little provenance guarantees for the input data. In this paper, we study adversarial ML with stock price forecasting. Adversarial incentives are clear and may be quantified experimentally through a simulated portfolio. We replicate an industry standard pipeline, which performs a sentiment analysis of Twitter data to forecast trends in stock prices. We show that an adversary can exploit the lack of provenance to indirectly use tweets to manipulate the model's perceived sentiment about a target company and in turn force the model to forecast price erroneously. Our attack is mounted at test time and does not modify the training data. Given past market anomalies, we conclude with a series of recommendations for the use of machine learning as input signal to trading algorithms.

## 1 INTRODUCTION

Research on the vulnerability of machine learning (ML) to adversarial examples (Biggio et al., 2013; Szegedy et al., 2013) focused, with few exceptions (Kurakin et al., 2016; Brown et al., 2017), on adversaries with immediate control over the inputs to an ML model. Yet, ML systems are often applied on large corpora of data collected from sources only partially under the control of adversaries. Recent advances in language modelling (Devlin et al., 2019; Brown et al., 2020) illustrate this well: they rely on training large architectures on unstructured corpora of text crawled from the Internet. This raises a natural question: *when the provenance of train or test inputs to ML systems is ill defined, does this advantage model developers or adversaries?* Here, by provenance we refer to a detailed history of the flow of information into a computer system (Muniswamy-Reddy et al.).

We study the example of such an ML system for stock price forecasting. In this application, ML predictions can both serve as inputs to algorithmic trading or to assist human traders. We choose the example of stock price forecasting because it involves several structured applications of ML, including sentiment analysis over a spectrum of public information sources (e.g., news, Twitter, etc.) with little provenance guarantees. There is also a long history of leveraging knowledge inaccessible to all market participants to gain an edge in predicting the prices of securities. Thales used his knowledge of astronomy to corner the market in olive-oil presses and generate a profit.

We first reproduce an ML pipeline for stock price prediction, inspired by practices common in the industry. We note that choosing the right time scale is of paramount importance. ML is better suited for low frequency intra-day and weekly trading than high-frequency trading, because the latter requires decision speeds much faster than achievable by ML hardware accelerators. Although there has been prior work on attacking ML for high-frequency trading (Goldblum et al., 2020), their experimental setting is *7 orders of magnitude* slower than NASDAQ timestamps (NASDAQ; 2020), which high-frequency trading firms use. In contrast, ML in low-frequency trading has attracted greater practical interest of the industry, with two major finance data vendors vastly expanding their sentiment data API offerings in the past decade (Bloomberg, 2017; Reuters, 2014). This is to serve

the growing demand from institutional and advanced retail market players to use ML models for sentiment analysis at lower frequencies.

We adopt this low frequency setting, and collect our data from Twitter and Yahoo finance, which provides 1-minute frequency stock price data. Using these services, we collected tweets related to Tesla and Tesla's stock prices over 3.5 years. We then used FinBERT (Araci, 2019), a financial sentiment classifier to extract sentiment-based features from every tweet. We describe these methods in more depth in Section 3.1; they are general and applicable to any company of interest.

We use sentiment features as our input and the change in price as our target to train multiple probabilistic forecasting models. We show that including sentiment features extracted from Twitter can reduce the mean absolute error of a model that only learns from historical stock prices by half. Section 3.2 introduces the probabilistic models used in our work. Predicting stock prices is a known hard task. Even a limited, per trade edge can lead to a large gain when scaled by the number of trades preformed (Laughlin, 2014). Moreover, typically sentiment analysis is only one of many indicators used in a trade decision. Hence our model only needs to provide a slight non-trivial advantage over random baselines to be effective in practice. We used our forecasts to implement portfolio strategies with positive returns, and measure its performance across other metrics to showcase its advantage.

Equipped with this realistic ML pipeline for stock price forecasting, which takes input data from a source with little provenance guarantees (i.e., Twitter), we set out to study its robustness. Unlike previous settings of adversarial examples like vision, where the realistic incentives for an adversary to change the classification is often unclear (Gilmer et al., 2018), in our setting there are clear financial interests at stake. Furthermore, Twitter is already subject to vast disinformation campaigns (Zannettou et al., 2019). These make it even more complicated to assess the provenance of data analyzed by ML pipelines. To investigate the robustness of our stock price prediction pipeline, we use adversarial examples to show that our financial sentiment analysis model is brittle in this setting.

While attacks against NLP models are not new in research settings, our work demonstrates the practical impacts that attacks on an NLP component (i.e., sentiment analysis) of a system can have on downstream tasks like stock price prediction. We show that an adversary can significantly manipulate the forecasted distribution's parameters in a targeted direction, while perturbing the sentiment features minimally **at test time only**. An adversary would determine a parameter and direction, such as increasing variance of forecasted stock prices, and compute a corresponding perturbation. If given control over training data, the adversary's capabilities would only further increase.

The contributions of this paper are the following:

- We propose a realistic setting for adversarial examples through the task of sentiment analysis for stock price forecasting. We develop a dataset collection pipeline using ticker data and Twitter discussion, for any ticker (i.e., company). This includes querying Twitter, processing data into a format fit for training, and a suitable sentiment analysis model.

- We implement different sentiment-based probabilistic forecasting models that are able to perform better then a naive approach in forecasting stock price changes given Twitter discussion surrounding the company. DeepAR-G, a Gaussian probabilistic forecasting model, outperformed all other models in our experiments.

- We subject our pipeline to adversarial manipulation, leveraging information from our model to minimally modify its inputs into adversarial examples, while achieving important changes in our model's output such as shifting distribution parameters of our probability distribution in any direction we wish.

We intend to release our code and data should the manuscript be accepted. Beyond the implications of our findings in settings where an adversary is present, we stress that capturing model performance in the worst case setting is important for the domain of finance. These threats are very real: past market anomalies led to the collapse of the Knight Capital Group, and there were legal matters following the suspected manipulation of Tesla stock via Twitter. Therefore, it is important for financial institutions to understand how their ML systems could behave in worst-case settings, lest market anomalies impact these systems in unprecedented ways. Furthermore, unforeseen catastrophic events (e.g., a natural disaster or pandemic) are often hard to model via standard testing procedures be it order generation simulators or backtesting, *i.e.* simulating trading on replayed past data. Our methodology based on adversarial examples enables institutional traders to assess maximum loss risk effectively.

## 2 BACKGROUND

**Stock Price Forecasting and Market Anomalies.** Market agents have long attempted to forecast stock prices. Nonetheless, predicting market behaviour is notoriously difficult for several reasons. If predictive factors become known, they alter the market's behaviour as agents attempt to exploit then. The limit of this behaviour is the efficient market hypothesis (French, 1970), which states that any information is immediately incorporated in prices, leaving a white noise process. The efficient market hypotheis has generated debate about whether it is possible to systematically outperform the market. See Cornell (2020) for a recent analysis suggesting that outperformance is possible, and may be linked to outsiders not understanding how such performance is attained. Lastly, markets follow the *Red queen effect* (Van Valen, 1973): as parties adapt their prediction capabilities, markets also evolve by becoming more challenging to forecast.

The deviations from the efficient market hypothesis are known as *market anomalies*. They occasionally allow individuals to capitalize off of such discrepancies. See Bass (1999) for a readable account of a successful application of chaos theory to market prediction. Beyond traditional techniques like arbitrage and standard data analysis, hopes for discovering factors that have not yet been traded away are presently centred on ML techniques, primarily, as the medium by which advances in computing power will drop the costs of research, including, for non-linear predictive factors. Thus, ML is a good fit to tackle the problem stated in the introduction. Indeed, ML models for tasks involving sequential data are now able to learn complex non-linear relationships. Initial work applied these advances to financial time series tasks (Fischer & Krauss, 2018).

**Sentiment Analysis in Finance.** Sentiment analysis serves as an input signal to trading decisions because it allows one to rapidly digest new information relevant to the market. This requires processing large quantities of text: e.g., news, financial statements, etc. Previous methods of extracting sentiment and moods were through curated lexicons where every word has a score associated with each sentiment (Bollen et al., 2011). Yet, word counting methods poorly reflect semantics since they are unable to factor in features like word order. Bollen et al. (2011) also received criticism for selecting a small and well specified testing set to achieve good results. On the other hand, training more complex models such as neural networks requires larger datasets. We follow the approach of Araci (2019) and turn to language models such as BERT (Devlin et al., 2019) or GTP-2 (Radford et al., 2019) finetuned on financial sentiment analysis datasets. Such models are able to factor in features that lexicon-based methods could not, achieving a deeper understanding of the text.

**Time Series Forecasting.** The general question of how to predict potential future trajectories from a series of observations is studied in the field of time series forecasting. While traditional modelling approaches like ARIMA (Box & Jenkins, 1968) and exponential smoothing (Hyndman et al., 2008) have focused on generating forecasts for individual time series, which are typically called *local* models, more recent techniques based on deep learning enable us to extract patterns from multiple (potentially related) time series jointly and capture these patterns in a *global* model. Popular architectures for time series forecasting include recurrent neural networks (RNNs) (Salinas et al., 2020; Smyl, 2020), convolutional neural networks (CNNs) (Oord et al., 2016; Borovykh et al., 2017; Bai et al., 2018) with 1-dimensional convolutions over time, as well as transformers (Vaswani et al., 2017; Lim et al., 2019; Li et al., 2019) using attention-based mechanisms. More details about the exact forecasting framework used in this work are established in Section 3.2.

**Adversarial Examples.** Extensive prior work in adversarial examples is in the vision domain (Szegedy et al., 2013; Su et al., 2019). Despite the fundamental advances in our understanding of the robustness of ML these works enabled, the practicality of such work is limited, and the incentives of the adversary to achieve such forms of misclassification is unclear (Gilmer et al., 2018). Recent work has focused on more practical domains, such as malware analysis (Kolosnjaji et al., 2018) or sentiment analysis of movie reviews (Samanta & Mehta, 2017). Studying the implications of adversarial examples in financial applications is clear: the financial gain will motivate malicious individuals to attack ML models deployed by financial institutions. This begs the question of whether these applications of machine learning to finance will exhibit the same lack of robustness than their counterparts did in the vision domain. In our work, we adapt the common strategy of determining the model's sensitivity to perturbations in the input through an analysis of the model's gradients. In particular, we leverage previous work on RNNs (Papernot et al., 2016).

## 3 METHODS

To achieve our goals of studying adversarial ML on a pipeline with little provenance guarantees, we perform the following. First, we collect a dataset comprised of a series of tweets about a particular organization and the associated stock prices. Next, we preprocess this data to obtain relevant features as needed by our predictive models (Section 3.1). For the models themselves, we utilize probabilistic forecasting approaches (Section 3.2). Finally, we show how these forecasting models can be attacked (Section 3.3). A detailed overview of the aforementioned approach is in Figure 1.

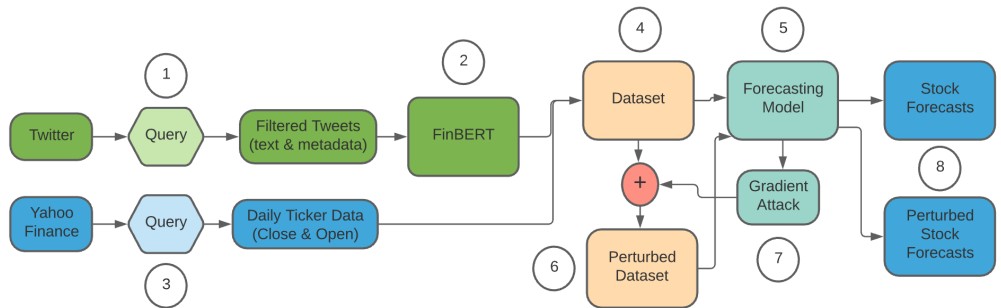

Figure 1: Tweets pertaining to a company were collected from Twitter (1) and passed through FinBERT, a financial sentiment classifier, to produce 10 sentiment based features (2). Price data from Yahoo finance was then collected to use as targets (3). We use the price data and sentiment features to create a dataset (4). We then train probabilistic forecasting models (5) on the collected dataset. Lastly, we adversarially manipulate Twitter based features in our testing set to obtain an adverserial testing set (6) using a gradient attack method (7). We then test the original and adversarial stock forecasts (8) using several metrics that simulate a portfolio's performance to determine the robustness of our model and data.

### 3.1 DATA COLLECTION

We now explain steps taken to create our dataset. Though we focus on one organization, our methodology is applicable to any organization of interest.

**Step 1. Collecting Tweets.** We chose to collect data about `Tesla` since it generates a large volume of discussion on Twitter. Furthermore, when collecting data for other companies, our querying method ran into 2 problems: 1. *homonyms:* it was hard to distinguish between Amazon the company and the Amazon rain-forest, and 2. *unintended discussion:* the majority of tweets pertaining to Microsoft were about Bill Gates being discussed along with other billionaires. Homonyms are particularly interesting as studies have shown that 12% to 25% of company pairs with similar names experience correlated price movements (Balashov & Nikiforov, 2019).

Tweets were collected by scraping Twitter posts that contain relevant words. This included: `Tesla`, `$TSLA`, `Elon Musk`, `Model 3` and more. Note that `$TSLA` is a special financial token called the company's *ticker*[1]. For every tweet, we collected (a) the timestamp, (b) the text associated with the tweet, and (c) the number of likes and retweets. In total, we collected 120,292 tweets spanning a 3.5 year period from 2017-01-02 to 2020-06-08. We limited ourselves to 896 days as the language used on Twitter frequently changes over time, introducing a temporal concept drift.

**Step 2. Collecting Stock Prices.** Ticker prices were collected via Yahoo Finance. Data for `Tesla` was collected for the same period as the tweets at a daily frequency, consisting of: (a) the opening price, and (b) the closing price. Opening price refers to the price when the market opens at 9:30 EST and closing price refers to the price when the market closes at 16:00 EST. Observe that tweets are continuous over time, while stock prices are not. Thus, we framed our time series data as follows: given tweets between 16:00 EST on day $t-1$ and 09:30 EST on day $t$, can we predict the difference between the opening price on day $t$ and closing price on day $t-1$. Lastly, it is common practice to predict the log of the ratio of open price to closing price, known as the *log return*.

---

[1]It is the symbolic name used to designate the stock on the exchange and look up price change information.

**Step 3. Performing Sentiment Analysis.** Whilst performing sentiment analysis on our dataset, we found that it is important to use a pretrained language model that was previously finetuned on financial data. We compared two datasets and approaches for finetuning BERT. The first approach involves finetuning a dense layer of BERT on the Sentiment140 dataset (Go et al., 2009), which contains 1.6M tweets labelled either positive or negative. The second approach involves using a finetuned BERT model on a corpus of financial sentiment data, called FinBERT (Araci, 2019). FinBERT classifies text into 3 categories: positive, negative, and neutral.

The first approach performed poorly in our tests. Although the second approach involves training on financial news and statements which contain a different distribution of vocabulary than those of tweets, we found that it performed significantly better on the data used to test the first approach. We thus used FinBERT as our sentiment classifier for the rest of our work. FinBERT outputs 3 logits: $o_-, o_+, o_=$ representing the confidence of negative, positive and neutral classes. The predicted sentiment class is the one with the highest logit, and Sentiment score is defined as $o_+ - o_-$.

**Step 4. Extracting Features.** Our dataset groups tweets into periods to match our daily ticker data. We explain the features we extract using information collected on September 17th 2019:

1. **Time Related:** (i) The date i.e., Tuesday, September 17 th 2019.
2. **Stock Related:** (ii) The opening price on Tuesday, September 17th 2019, and (iii) the closing price on Monday, September 16th 2019.
3. **Tweet Related:** (iv) Number of tweets, (v) average likes, and (vi) average retweets.
4. **Sentiment Based:** (vii,viii,ix) The percentage of tweets belonging to each sentiment class, (x) average sentiment score , and (xi,xii,xiii) average of each class's logit.

This results in 7 data points derived from sentiment and 3 from tweet metadata, resulting in 10 features. Additionally, we also utilize the date index, and the two stock related data points to calculate the log return of the target stock. Although the markets are closed over the weekend, events pertaining to or influencing the stock (e.g., an earthquake close to one of the company's factories) can occur over this period and be discussed on Twitter. Thus, Monday morning data points contain Twitter discussion from Friday after 16:00, the entire weekend, and up to 9:30 on Monday.

## 3.2 PROBABILISTIC FORECASTING

Using the dataset collected, we can learn a global model over the complete panel of time series to predict the stock's log return. Rather than outputting a point-forecast for these values, we use a probabilistic forecasting strategy which allows us to account for the inherent uncertainties in the prediction. As part of our setup, we investigate both the usage of a univariate time series model, which forecasts each time series independently of the other time series, and the multivariate setting in which we produce forecasts over multiple time series jointly.

The univariate probabilistic forecasting setting can be summarized as follows. Given a set $Z = \{z_{i,1:T_i}\}_{i=1}^N$ of univariate time series, with each element $z_{i,1:T_i} = (z_{i,1}, z_{i,2}, \ldots, z_{i,T_i})$, and a set of associated covariate vectors $X = \{\boldsymbol{x}_{i,1:T_i+\tau}\}_{i=1}^N$, we intend to model the probability distribution over future (unobserved) values $z_{i,T_i+1:T_i+\tau}$ of length $\tau$ conditioned on past time series values $z_{i,1:T_i}$ and covariates $\boldsymbol{x}_{i,1:T_i+\tau}$ using a neural network $\mathcal{M}$ parameterized by $\Phi$:

$$p(z_{i,T_i+1:T_i+\tau} \mid z_{i,1:T_i}, \boldsymbol{x}_{i,1:T_i+\tau}) = \mathcal{M}_\Phi(z_{i,1:T_i}, \boldsymbol{x}_{i,1:T_i+\tau})$$

Note that covariates are assumed to be known for the full prediction length $\tau$. This setup can be generalized to multivariate forecasting over $D$-many time series by replacing predictions for a single time series $z_{i,t} \in \mathbb{R}$ with predictions over a vector of time series $\boldsymbol{z}_t \in \mathbb{R}^D$.

As part of this work we use the DeepAR (Salinas et al., 2020) (see Figure 8 for an overview) and DeepVAR (Salinas et al., 2019) architectures, which are autoregressive RNNs specifically designed for time series forecasting in the univariate and multivariate setting respectively. Specifically, we use their Python implementations in GluonTS[2] (Alexandrov et al., 2019) for our experiments. Both models allow for flexible parametric and non-parametric output distributions by using a projection

---

[2]https://github.com/awslabs/gluon-ts

layer to map the RNN output to parameters defining the chosen distribution function. For example, if the chosen distribution is Gaussian, DeepAR parameterizes the mean and standard deviation at each time step $t = 1, \ldots, T_i + \tau$. Finally, we also consider the GPVAR model (Salinas et al., 2019) which parameterizes the output as a low-rank Gaussian Copula process.

When working with DeepAR, we consider stock log returns as our single time series and the Twitter features as covariates. Recall that covariates are assumed to be known over the entire time period under consideration, including the prediction horizon of length $\tau$. This is not the case with Twitter data as we do not know future tweets. To alleviate this issue, we consider forecasting 1 day in the future or $\tau = 1$. This is analogous to where a trader would collect all tweets posted after the previous day's market closure, and predict the new day's opening price. When working with DeepAR's multivariate version DeepVAR and GPVAR, we consider the stock log returns and our sentiment features as time series we wish to predict. In this setting, we have no restriction on $\tau$, however the model is learning distributions for other series as well. When training, we cannot translate a decreasing loss to improvements towards a specific time series.

## 3.3 GRADIENT ATTACK

The crafting method of our adversarial sequences is similar to Papernot et al. (2016). In their implementation, to perturb output $j$, they alter input $i$ if output $j$ is highly sensitive to input $i$, and other outputs $k \neq j$ are not sensitive to input $i$.

We implemented a gradient attack for DeepAR alone, but the idea stays the same for multivariate settings. The inputs of our model are our covariates features $\boldsymbol{x}_{i,1:T_i+\tau}$, while our outputs are the set of parameters defining our probability distribution, $\Theta = \{\theta^1, \ldots, \theta^K\}$. For each time step, we calculate the gradient of our distribution parameters with respect to each covariate, resulting in a 3 dimensional tensor of shape $(\tau, N, K)$ for our Jacobian. The value $J[t, a, k]$ represents how sensitive parameter distribution $\theta^k$ is at time step $T_i + t$, denoted $\theta^k_{T_i+t}$, with respect to covariate $\boldsymbol{x}_{a,T_i+t}$.

---

**Algorithm 1** Adversarial sequence crafting: Given 1 distribution parameter $\theta^*$ of the distribution that is defined by the set of distribution parameters $\Theta$ that we wish to push in the direction $d$, we iteratively perturb covariate $x_{a^*,T_{a^*}+t^*}$ by a fixed $\delta$ if $\theta^*_{T_{a^*}+t^*}$ is highly sensitive to that point and other distribution parameters are not.

**Require:** $\Theta, \theta_*, \mathbf{x}_{i,1:T_i+\tau}, d, \delta, R$
1: $\mathbf{x}^* \leftarrow \mathbf{x}$
2: **for** $j = 0, 1, \ldots, R$ **do**
3: $\quad a^*, t^* = \underset{a,t}{\operatorname{argmax}} |J[t, a, *]| \times \left| \sum_{\theta^k \in \Theta \setminus \theta^*} J[t, a, k]) \right|^{-1}$
4: $\quad \mathbf{x}_{a^*,T_{a^*}+t^*} \leftarrow \mathbf{x}_{a^*,T_{a^*}+t^*} + \delta * (d * sgn(J[t^*, a^*, *]))$
5: **end for**
**return** $x^*$

---

The input to our crafting algorithm is a distribution parameter we wish to perturb, $\theta^*$, a direction $d$ which is the direction in which we wish to perturb the input to. $d$ is -1 if we wish to decrease $\theta^*$, or +1 if we wish to increase $\theta^*$. We determine the covariate $a^*$ and time $t^*$ which would perturb $\theta^*_{T_i+t^*}$ significantly, while perturbing other distribution parameters minimally. We do so by computing the ratio of $|J[t, a, *]|$ to $\left| \sum_{\theta^k \in \Theta \setminus \theta^*} J[t, a, k] \right|$ for each time step $t$ and covariate $a$, and we pick the $t$ and $a$ that maximize this ratio. We then perturb $\mathbf{x}_{a^*,T_{a^*}+t^*}$ by a fixed value $\delta$ in the direction $d$. For example, if our algorithm picks the positive sentiment score at time $t$ and increases it by $\delta$, the adversary may: (1) post a few tweets with strong positive sentiment or (2) post many tweets that are slightly positive. The algorithm devised in Papernot et al. (2016) for misclassifying movie reviews can be used for this task by targeting FinBERT instead. The adversary could start with random tweets or tweets posted in the past and perturb them until they reach the require positive score.

Lastly, the setting in Papernot et al. (2016) is classification of sentences, and the input is perturbed until the sentence is misclassified. As our task is not classification we do not have such a condition. Instead, we perturb the input for a fixed number of steps $R$.

# 4  EVALUATION

## 4.1  TO WHAT EXTENT CAN WE FORECAST STOCK PRICES WITH THE DESCRIBED PIPELINE?

Here we study whether our stock forecasting models are capable of producing meaningful forecasts before we attempt to attack them, and if so, quantify their usefulness.

We run our 3 models under different parameters and compare their results across 11 metrics shown in Table 1. In particular, we evaluate the potential financial gains that a trader can expect through the Sharpe ratio and two active strategies: Greedy and Threshold. An explanation of these metrics is found in Appendix B.1, as well as the hyper-parameters used (refer Appendix B.2.1). We used a 90-10 training testing split, resulting into a testing set of 89 days shown in Figure 7. As we are interested in predictions lengths $\tau \in \{1, 3\}$, we implement a rolling window over our testing set. Instead of plotting probabilistic forecasts that are seperated due to our rolling testing set such as Figure 3, we plot the mean of 100 samples paths taken from our distribution as in Figure 2.

| Model | MAE ($\times 10^{-2}$) | MAPE | MASE | RMSD ($\times 10^{-7}$) | CRPS | Accuracy | Sharpe Ratio | Greedy | Threshold |
|---|---|---|---|---|---|---|---|---|---|
| DeepAR-S-ST | 6.45 | 0.0155 | 1.449 | 2.127 | 1.265 | 49.45% | -31.9% | $ 4068 | $ **7605** |
| DeepAR-S-G | 5.00 | 0.0199 | 1.110 | 0.670 | 1.381 | 50.55% | -26.1% | $ **4553** | $ 7352 |
| DeepAR-ST | 5.93 | 0.0139 | 1.548 | 2.639 | 1.230 | 57.14% | -2.23% | - $ 400.8 | $ 6152 |
| DeepAR-G | **3.20** | **0.0109** | **0.996** | **0.524** | **1.199** | **57.14%** | **3.31%** | $ 502.1 | $ 6103 |
| DeepVAR | 5.37 | 0.0170 | 1.592 | 4.666 | N/A | 51.65% | 0.25% | - $ 1227 | $ 5575 |
| DeepVAR-3 | 5.61 | 0.0145 | 1.356 | 1.698 | N/A | 54.95% | 0.20% | -$ 107.8 | $ 5440 |
| GPVAR | 5.59 | 0.0135 | 1.262 | 1.263 | N/A | 52.75% | -0.17% | $ 364.2 | $ 6584 |
| GPVAR-3 | 4.61 | 0.0126 | 1.198 | 0.771 | N/A | 47.31% | -0.05% | - $ 216.5 | $ 6422 |

Table 1: Different forecasting models on a 90 day rolling window testing set. The target has a binary distribution of 54.95%, Sharpe ratio of 9.04% and passive gain of $ 245.3. DeepAR-G is the best performing model across every metric except our portfolio methods. CRPS values are not available for multivariate models as they were aggregated for all time series.

Our first experiment is to determine the usefulness of Twitter covariates in a univariate setting. We trained DeepAR without Twitter covariates on Gaussian and Student-t distributions, and then include Twitter covariates on the same 2 distributions. For our purposes, the important difference between the two distributions is that Student-t has heavier tails than the Gaussian distribution, resulting in a higher likelihood of extreme behaviour. These 4 scenarios are the 4 first rows shown in Table 1, where $-S-$ indicates removing covariates, no $-S-$ indicates the use of covariates and lastly $-ST$ and $-G$ indicate Student-t and Gaussian distributions, respectively.

Including Twitter features as covariates improved performance across all metrics for Gaussian distributions except the two portfolio metrics. Furthermore, the portfolios gains are only slightly worse and still a positive return. Student-t performs worst across all implementations with the Twitter covariates. Figure 2 shows the forecasts of DeepAR-ST, and despite its performance, the shape of the forecasted log return is consistent with the real change in stock, but shifted horizontally or vertically by a couple days or percentage log returns. These shifts make the overall performance of this method very poor, although the model has been able to learn something about the true forecasts.

Our second experiment tests our multivariate models such as DeepVAR and GPVAR. We ran 2 scenarios for each, one with a prediction length of 1 and another with a prediction length of 3. We found that all 4 implementations did not perform as well as the univariate models. We show the distribution output of GPVAR-3 in Figure 3. In this setting, the model is trained to learn a joint distribution over all 11 time series. We hypothesize that when the loss is being optimized, the Twitter-based time series are significantly easier to learn compared to the log return of Tesla stock. Hence although our model's loss converges, very little improvement has been done in forecasting the log return of Tesla stock compared to the Twitter-based features.

Hence in Table 1, the best implementation was a Gaussian distribution using Twitter covariates–DeepAR-G. We visualize the resulting forecast in Figure 4. We now determine the performance of our model not relative to each other, but in an absolute sense. Although the financial metrics show success with positive returns, this is likely due to the overall upward trend Tesla had during our testing set. The metric that is best suited for this task is MASE, which takes the ratio of the error of our forecast, to the error associated with predicting the previous days forecast. In every case except DeepAR-G, this value is above 1, indicating that predicting the previous day is better. In DeepAR-

G, the MASE is 0.996, indicating that the forecasted method is just slightly better then predicting the previous day's change. Accompanied by the discussion regarding DeepAR-ST, we believe that we were able to implement a pipeline that has limited but *non-trivial* forecasting ability.

## 4.2 TO WHAT EXTENT IS OUR MODEL AND DATA VULNERABLE TO MANIPULATION?

We implement the attack formulated in Section 3.3 to both of our best performing models, DeepAR-G and DeepAR-ST, for different distribution parameters and directions. We characterize how much perturbation in our features is required to change our models' forecasts. We summarize our results in Table 2 and Figures 5 and 6, where we show that an adversary is able to manipulate a distribution parameter in a direction of their choosing to reduce the performance of forecasted predictions.

| Distribution | Param | Dir | MAE ($\times 10^{-2}$) | MAPE | MASE | RMSD ($\times 10^{-7}$) | CRPS | Accuracy | Sharpe | Greedy | Threshold |
|---|---|---|---|---|---|---|---|---|---|---|---|
| Student-t | – | – | 5.93 | 1.39% | 1.548 | 2.639 | 1.230 | 57.14% | -2.23% | - 400 | 6152 |
| Student-t | $\mu$ | ↑ | 11.53 | 2.48% | 2.323 | 11.863 | 1.545 | 54.95% | -31.76% | 5911 | 7658 |
| Student-t | $\mu$ | ↓ | 10.72 | 1.99% | 1.861 | 4.805 | 1.7131 | 56.04% | -0.27% | 3067 | 6053 |
| Student-t | $\nu$ | ↑ | 6.01 | 1.87% | 1.753 | 4.571 | 1.8071 | 58.24% | -0.56% | 11618 | 7625 |
| Student-t | $\nu$ | ↓ | 7.68 | 1.76% | 1.648 | 2.967 | 1.3951 | 58.24% | 14.7% | 1803 | 8602 |
| Student-t | $\sigma$ | ↑ | 7.67 | 1.76% | 1.647 | 2.963 | 1.8109 | 57.14% | 14.8% | 1732 | 7005 |
| Student-t | $\sigma$ | ↓ | 8.31 | 1.80% | 1.688 | 3.93 | 2.4318 | 54.95% | 4.89% | 1803 | 6802 |
| Gaussian | – | – | 3.20 | 1.09% | 0.996 | 0.524 | 1.199 | 57.14% | 3.31% | 502 | 6103 |
| Gaussian | $\mu$ | ↑ | 10.76 | 2.59% | 2.425 | 14.97 | 1.8874 | 58.24% | 55.59% | 402 | 5964 |
| Gaussian | $\mu$ | ↓ | 10.31 | 1.94% | 1.815 | 4.961 | 2.3299 | 54.95% | 54.12% | 10220 | 9818 |
| Gaussian | $\sigma$ | ↑ | 10.32 | 1.94% | 1.815 | 4.963 | 2.3303 | 59.34% | 57.07% | 332 | 5964 |
| Gaussian | $\sigma$ | ↓ | 8.131 | 1.55% | 1.449 | 1.704 | 1.996 | 52.75% | 17.11% | 224 | 6647 |

Table 2: Performance of crafting algorithm on different distributions and directions $d$. Every type of perturbation resulted into larger error over all error metrics. Note that some experiments resulted in similar performance such as $\nu \downarrow$ and Student-t $\sigma \uparrow$, likely due to the same perturbations.

Recall that the inputs to our attack are (a) the distribution parameter $\theta^*$, and (b) a direction $d$ along which $\theta^*$ should be perturbed. In Table 2, we manipulate every distribution parameter of a Gaussian and Student-t distribution in both directions. We also report the unperturbed performance for comparison. The performance of adversarially manipulated forecasts across every experiment in Table 2 is worse then the original forecasts. Other metrics such as binary accuracy, Sharpe ratio, and our portfolio metrics (see Appendix B.1) show mixed results. The reason is in our probabilistic setting perturbing a distribution parameter does not always decrease the strength of forecasts. For example, Figure 6 shows the result of decreasing the mean. Although we achieved the intended change in mean by going from $2.109 \times 10^{-4}$ to $-9.57 \times 10^{-3}$, at day 05-01 in Figure 6, our adversarial forecast is better than our original forecast. The original forecast was above the target, and thus decreasing the mean actually *improved our forecasts*. Hence, in the application of Algorithm 1, an adversarial trader would need to use knowledge about the victim model's forecast to determine the distribution parameters, direction, and number of iterations to achieve their goal.

Figure 5 shows the perturbed features from trying to decrease the mean of DeepAR-G, while Figure 6 plots the true, forecasted and adversarially forecasted log return of Tesla stock. We perturbed our Twitter covariates using algorithm 1 for $R = 30$ iterations resulting into the original and perturbed features in Figure 5: we can see small perturbations in our covariates represented by dashed lines. For example, covariate 4 from the bottom is perturbed at around day 13 and 32, while the 7th and 8th covariates from the bottom are perturbed significantly at day 82 of our dataset. In this scenario, we picked a value of $\delta$ that is 1% of the range of the covariate across the whole dataset. The adversarially forecasted log returns are on average smaller as described in the previous paragraph, which is consistent with the intended attack goal of decreasing the mean of our forecast.

**Conclusions.** We demonstrated how the realistic stock price forecasts of our pipeline, initially able to turn a profit through active trading, are subject to worst-case drops in predictive performance, e.g., when an adversary manipulates inputs from low provenance data pipelines. Our methodology helps appreciate maximum loss risk faced when incorporating ML in trading. Future work will need to integrate outlier detection for sentiment analysis inputs, and integrate these detection results with probabilistic forecasting to increase robustness of the end-to-end pipeline to worst-case inputs.

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

APPENDIX

## A    PLOTS

Figure 2, 3 and 4 shows the resulting forecast of DeepAR-ST, GPVAR-3 and DeepAR-G, respectively over our testing set of 89 days. Their performance is shown in Table 1 and discussed in Section 4.1. Figure 5 plots the original and perturbed testing sets resulting into the original and adversarial forecast in Figure 6, respectively. The perturbation was for decreasing the mean of DeepAR-G. Lastly, Figure 7 shows Tesla's log return over the training and testing set, which we discuss in Appendix B.4.

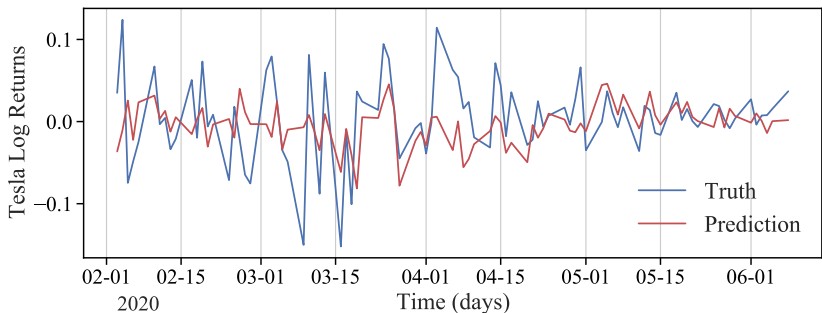

Figure 2: Log return of Tesla stock over our testing set for DeepAR-ST. Note the similar but shifted shape of our prediction and the target.

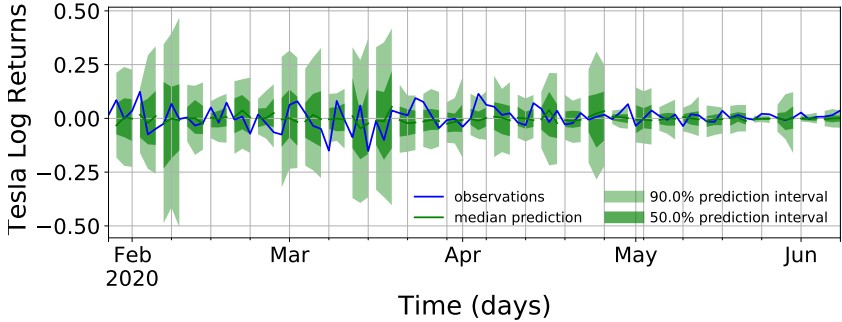

Figure 3: Probability distribution of log return of Tesla stock over our testing set for GPVAR-3. The faint green line within the probability distribution represents the mean prediction of 100 samples of our distribution.

## B    METRICS

### B.1    PERFORMANCE METRICS

We include information regarding the performance metrics used throughout Section 4. We divide our metrics into error and financial metrics shown below. In all settings, we consider the true log return of Tesla stocks at time $t$ as $y_t$ and the predicted as $\hat{y}_t$ over a testing set of $T$ days.

### B.1.1    ERROR METRICS

The error metrics all use the difference between our predictions and the truth in different representations. MAE describes the mean absolute error while MAPE describes the average percentage error, shown in equations 1 & 2, respectively. Both metrics capture how close our forecasts are on average, however this fails to account for direction. We prefer to have predictions and targets of the same

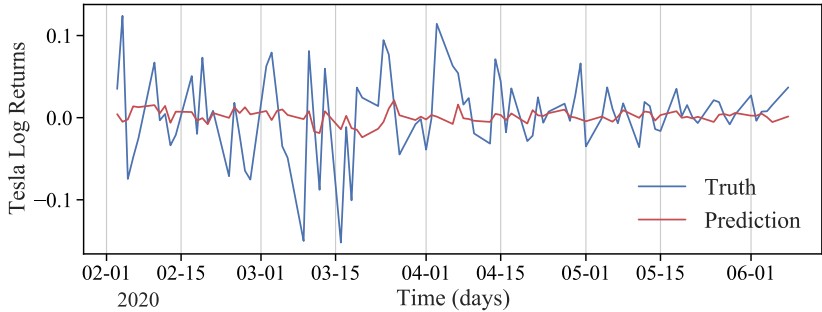

Figure 4: Log return of Tesla stock over our testing set for DeepAR-G. Compared to DeepAR-ST, the predictions have a lower variance due to Gaussian distribution having smaller tails then Student-t distribution.

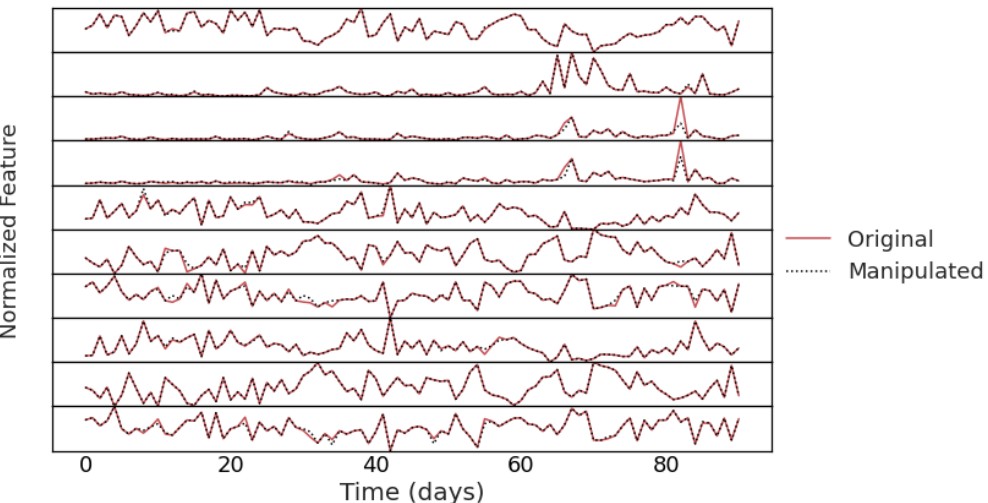

Figure 5: Plotting the normal and adversarial versions of the covariates used in the testing set. From top to bottom, the plots represent the following covariates: general sentiment score, volume of tweets, average likes, average retweets, positive sentiment score, negative sentiment score, neutral sentiment score, percentage of positive tweets, percentage of negative tweets and percentage of neutral tweets. The goal of the perturbation here was to decrease the mean of DeepAR-G. Note the perturbations of neutral score during between day 10 to 35, or average retweets at around day 65 or 82.

sign and differ in magnitude greatly, rather then a small difference in magnitude with differing signs. To circumvent this problem, we report the binary accuracy of our model shown in equation 5. This measures the percentage of forecasts that matched the targets direction, regardless of magnitude. In our testing set, we had an underlying distribution of 54.95% positive true log returns.

Mean absolute scaled error, MASE, is a ratio of MAE to the MAE of using yesterday's true log return as today's forecast. This metric is interesting as a value below 1 shows that our model outperforms the idea that the market is a random process and the best forecast for the future is the present, also known as a Martingale. Root mean squared deviation uses takes the square root of the average squared error. This metric is useful for outliers: a single forecast that is very far off the true target increases the RMSD significantly more then many forecasts that are slightly off. Lastly, the cumulative ranked probability score, CRPS, is a generalized MAE for probabilistic forecasting, and one of the most widely used accuracy metrics for probabilistic forecasting.

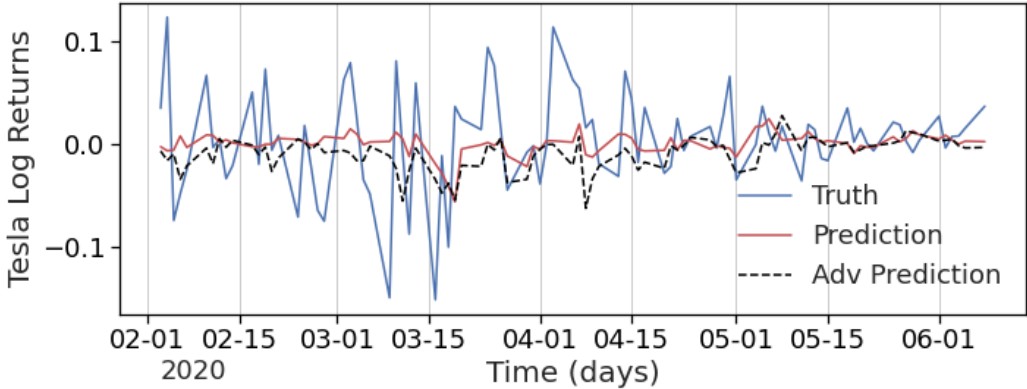

Figure 6: Change in predictions after adversarially decreasing the mean of DeepAR-G.

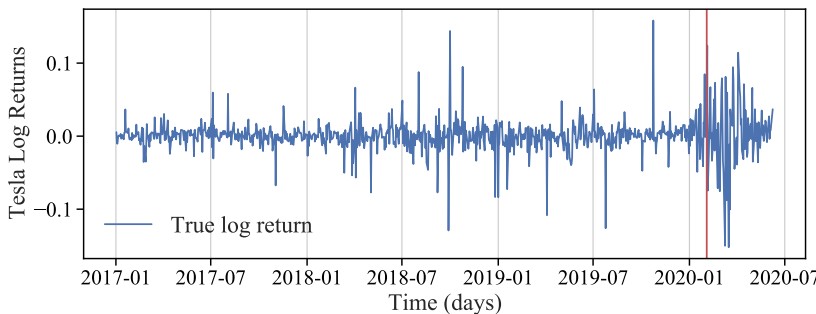

Figure 7: Log return of Tesla stock across our whole dataset. A red vertical line separates the training and testing sets. Note the difference in average magnitude of log returns in the training set compared to the testing set.

$$\text{MAE} = \frac{1}{T} \sum_{t=1}^{T} |y_t - \hat{y}_t| \tag{1}$$

$$\text{MAPE} = \frac{1}{T} \sum_{t=1}^{T} \left| \frac{y_t - \hat{y}_t}{y_t} \right| \tag{2}$$

$$\text{MASE} = \frac{MAE}{\frac{1}{T-1} \sum_{t=2}^{T} |y_t - y_{t-1}|} \tag{3}$$

$$\text{RMSD} = \left( \frac{1}{T} \sum_{t=1}^{T} (y_t - \hat{y}_t)^2 \right)^{\frac{1}{2}} \tag{4}$$

$$\text{Binary Accuracy} = \frac{1}{T} \left( \sum_{t=1}^{T} \mathbb{1}[sgn(\hat{y}_t) = sgn(y)] \right) \tag{5}$$

### B.1.2 FINANCIAL METRICS

While machine learning metrics are important to evaluate our model, the best evaluation that we can propose for a stock price prediction algorithm is always the potential financial gains that a trader can expect.

**Sharpe ratio**  Since trading is inherently risky, we will need to evaluate the return of our strategy adjusted for the extra risk that we are taking by running this strategy over not doing anything. The *Sharpe ratio* is a standard financial metric designed specifically for this purpose. Sharpe $= (r_{\text{strategy}} - r_{\text{risk free}})/\sigma$ with $\sigma$ the standard deviation deviation of the returns $r_{\text{strategy}}$ For simplicity, we assume that by doing nothing we gain nothing, *i.e.* $r_{\text{risk free}} = 0$.

**Trading strategies**  We introduce a baseline strategy and 2 simple active trading strategies that act as a toy example of how a real market participant would use the edge gained by our price prediction model. To be consistent with our low frequency setting, we allow at most one trade each day.

For these strategies, the low frequency day by day trading approach enables to make a few first order simplifying averaging assumptions that would not be true in the HFT setting. We assume that orders are executed immediately at the price for that day. Moreover, in this setting we do not have to assume crossing the spread on each trade, a major hurdle of active strategies that may eat in the profit reported by backtesting. Trading fees paid to the exchange are assumed to be zero.

Our baseline, which we call *Passive Gain*, is to buy one share first day of the period and sells it on the last day. This simple strategy is an adequate baseline as it represents *inventory risk*, that is the change in portfolio value not related to active trading, faced by specific participants such as *institutional market makers* (Guéant et al., 2012).

Our first active strategy, called *Greedy Gain*, blindly follows the stock price prediction. If the model predicts a price increase the trader, human or algorithm, buys a share. If the model predicts a drop, the trader sells a share if he holds at least one. This forgetful strategy exposes the trader to the maximum amount of risk from active trading.

Our last strategy, called *Threshold Gain*, keeps a history of past $k$ Opening prices. In this stateful strategy, the trader still only uses the stock price directional prediction. However, they only buy or sell if the prediction is more than one standard deviation away from the mean price over the history. If the prediction is closer than one standard deviation from the mean, they do nothing. While this strategy involves smaller risk exposure than the Greedy Gain one, the trader may at times withhold from trades which would have been profitable.

## B.2  Training and Adversarial Crafting with DeepAR

### B.2.1  Training

Except prediction length and number of epochs, the hyper-parameters of DeepAR, DeepVAR, GP-VAR and the mxnet trainer were the default hyperparameters, described in their documentation[3]. We used 25 epochs for DeepAR, 40 epochs for DeepVAR, and 50 epochs for GPVAR respectively. For all 3 models, Twitter based features were standardized to $\mu = 0$ and $\sigma^2 = 1$. Figure 8 shows the DeepAR architecture from the original work in depth.

## B.3  Adversarial Crafting

In our adversarial crafting method, we did not wish to manipulate a certain feature to become the new maximum or minimum across the whole dataset. If our perturbation pushes our feature to become the new maximum or minimum, we pick the covariate and time step that satisfies our argmax that is not the previous candidate covariate and time step, and perturb it instead. For the perturbation amount $\delta$, we tried a fixed size perturbation, a scaling factor and a variable sized $\delta$ for each covariate. All 3 methods were able to achieve the intended results of our attack, but the extent and shape of the perturbed features in Figure 5 varied across the 3 methods.

## B.4  Training and Test Distribution

In Figure 7, the training set has occasional large log returns, but these are infrequent and spaced apart. The testing set however has large price changes, and occur one after the other. This period lines up with the beginning of COVID-19, where many companies including Tesla experienced large volatility. We found Tesla tweets collected during periods of great volatility contained more tweets

---
[3]https://github.com/awslabs/gluon-ts

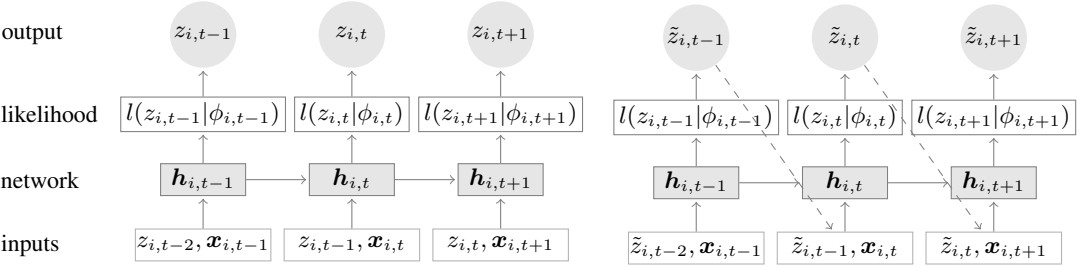

Figure 8: Overview of the DeepAR model (figure taken from Salinas et al. (2020)). Inputs $z_{i,t-1}$ and $\boldsymbol{x}_{i,t}$ as well as the previous RNN hidden state $\boldsymbol{h}_{i,t-1}$ are fed to the RNN's current state to compute $\boldsymbol{h}_{i,t}$ for each time step $t$. The RNN's output is then mapped to the parameters $\phi_{i,t}$ governing the likelihood function $l(z_{i,t}|\phi_{i,t})$ associated with a specific distributional assumption over $z_{i,t}$. Training is depicted on the left for which we require $z_{i,t}$ to be known; autoregressive prediction is shown on the right where a sample $\tilde{z}_{i,t} \sim l(\cdot|\phi_{i,t})$ is drawn from the predictive distribution at $t$ and fed back into the prediction for $t+1$.

discussing financial events related to Tesla, such as plans for opening factories or new products, rather then opinions surrounding Elon Musk or Tesla. We believe that a stronger financial distribution of tweets made the sentiment derived from FinBERT a strong signal compared to periods in the training set. We experimented with reducing the training test split to smoothen out the different distributions across sets, however the model performance suffered. We believe that this is could be because deep neural networks such as DeepAR requires enough data points to learn a meaningful relationship especially a complicated one such as the log returns of a stock. Another reason would be that we did not catch financial signal from Twitter during periods with less variance and hence a data source and collection issue.

