# OpenReview forum: "On the Robustness of Sentiment Analysis for Stock Price Forecasting"
_ICLR.cc/2021/Conference — Reject_

### Official Review · AnonReviewer1 · 2020-10-25
**An interesting topic but several issues in experiments**

**Rating:** 5
**Confidence:** 4

**Review:**

The authors in this paper propose to use the adversarial ML in the stock price forecasting scenario and demonstrate that an adversary can exploit the lack of provenance to indirectly use tweets to manipulate the prediction model’s perceived sentiment about a target company and in turn force the model to forecast price erroneously. However, I have the following comments on the paper:

- Dataset collection is not convinced. Authors collect only one company (i.e.,  Tesla) data from Twitter and Yahoo. Experiments in the entire paper are only based on the dataset containing just one company. The authors provide two reasons for not considering other companies (such as Microsoft or Amazon as discussed examples in the paper): "homonyms" and "unintended discussion". However, Tesla also has the similar issue. For example, when Elon Must was discussed on Twitter, it may be about Elon Musk’s other companies (SpaceX) or his arguments that are not related to Tesla (similar to the Bill Gates case in the paper). When people talk about Tesla, they may talk about Nikola Tesla. When people discuss Model 3, they may only express their experience of driving Model 3. The dataset with only one company contains bias to the model that may not be generalized to other cases.

- Lack of novelty. All the technical components in this paper are not new. FinBERT, DeepAR, and Gradient Attack are all from existing works.

---

> ### Author Response · Authors · 2020-11-24
> **Goal of our work & additional dataset comments**
>
> We thank the author for their comments and insight.
>
> The goal of our paper was to develop a stock price forecasting pipeline and showcase the capabilities of an adversarial ML attack to the pipeline. We decided to investigate a single company as a proof of concept. We decided to work with Tesla in large part because of the significant volume of discussion on Twitter and Elon’s Musk frequent use of Twitter. Although we agree with the reviewer that our work can be improved by testing other datasets, the purpose of our work is not to create a state of the art forecasting model for any company, rather showcase adversarial examples in a more meaningful setting compared to previous domains (movie reviews, image classification etc).
>
> Nonetheless, we tried to collect tweets from another company, however Twitter updated their API at the end of September, breaking our scraping tool (and many others). Unfortunately, this severely limits the range of data we can collect (only a month of data) until the review stage. This is not enough data to train any of our models.
>
> Regarding the novelty of our work, the goal of this paper is to showcase the threat vector that may exist when using machine learning models on public data via adversarial crafting, not to showcase a novel pipeline or attack method. Implementing the pipeline used in our evaluation required non-trivial efforts which we hope will lay the foundations for others to study adversarial examples in a domain where adversarial incentives are clear. We hope our work serves as a benchmark for much more work within adversarial machine learning in high-risk domains such as finance, hence we believe that the value of our effort stems in the non-trivial forecasting model implementation combined with a dedicated robustness analysis
>
> Lastly, the conclusion of our work is not as simple as the reviewer mentioned. The inputs are not perturbed randomly, rather they are perturbed such that the forecasts are changed following an adversary’s objective. The adversary selects a distribution parameter and direction, and the algorithm returns a set of perturbations to achieve such a goal. For example, an adversary could wish to decrease the mean forecast by a fixed amount, and the algorithm would return a number of tweets to post with a specific negative score, along with the number of likes and retweets for each of them.

---

### Official Review · AnonReviewer4 · 2020-10-26
**On the Robustness of Sentiment Analysis for Stock Price Forecasting**

**Rating:** 7
**Confidence:** 4

**Review:**

Pros:
The paper is clear with a significant contribution. It performed a sentiment analysis task that can predict trends in the stock market and also showed how an adversary may attack the model using tweets thereby leading to false price prediction. The methodology and the probabilistic forecasting used are excellent in my opinion.

Cons:
1. The title of the paper does not describe the actual work done. I suggest that the authors may consider giving a new title to reflect the method they applied.
2. It is unclear why the attack was done on the test stage but was not shown at implementation. That is, how would this work be reproduced or be beneficial to the community?

The following can be corrected:
1. The claim on Page 1, Para 2 of the Introduction needs a citation (Thales used his...)
2. On page 5, para 1, please briefly describe BERT

---

> ### Author Response · Authors · 2020-11-24
> **Changing the title**
>
> We thank the reviewer for their comments. We appreciate that the reviewer finds this as rewarding as ourselves.
>
> We agree with the reviewer about the title of our work. We are thinking of another title to better reflect our methods.
> Regarding point #2, We are unsure what the reviewer meant by “not shown at implementation”. We are willing to open-source all of our code and data collected upon publication of the work. We hope that our pipeline and data collections tools can be used as a baseline for further work in the underlying and combating the threats associated with machine learning in high-risk industrial applications like finance.

---

### Official Review · AnonReviewer2 · 2020-10-27
**Official Blind Reviewer 2**

**Rating:** 5
**Confidence:** 3

**Review:**

In this paper, the authors studied the problem of adversarial ML in stock price forecasting. They first replicated an industry standard pipeline, which performs a sentiment analysis of Twitter data to forecast trends in stock prices. Then, they show that an adversary can exploit the lack of provenance to indirectly use tweets to manipulate the model’s perceived sentiment about a target company and in turn force the model to forecast price erroneously.

Strength:

1. The topic is interesting. Adversarial ML attracts a lot of attention recently, and applying them to stock price forecasting has clear incentive.
2. The authors plan to release the code and data when this is accepted, which is good for other researchers to reproduce this work.

Weakness:

1. The novelty of the methodology seems limited. Both the stock price forecasting model (FinBERT and DeepAR-G) and the gradient attack method (Papernot et al. 2016) are not proposed by the authors.
2. The writing can be improved. For example, in the experiment section, the authors used a lot of evaluation metrics such as sharpe ratio, greedy, threshold. Though they might be familiar to the audience in finance industry, they are not very clear to me at first. I'd suggest to give a definition and explanation of those metrics in this section.

Overall comments:

I think the research topic of this paper is interesting, but the methodology seems lack of novelty. In addition, the writing can be further improved.

---

> ### Author Response · Authors · 2020-11-24
> **Scope of our work**
>
> We thank the reviewer for their review and time.
>
> The goal of this paper is to showcase the threat vector that may exist when using machine learning models on public data via adversarial crafting, not to showcase a novel pipeline or attack method. Implementing the pipeline used in our evaluation required non-trivial efforts which we hope will lay the foundations for others to study adversarial examples in a domain where adversarial incentives are clear. We hope our work serves as a benchmark for much more work within adversarial machine learning in high-risk domains such as finance, hence we believe that the value of our effort stems in the non-trivial forecasting model implementation combined with a dedicated robustness analysis. Regarding our evaluation metrics, we described every metric in the appendix, but can be introduced in the body of our work and reference readers to the appendix for more detailed information.

---

### Official Review · AnonReviewer3 · 2020-10-28
**This is generally an interesting paper as it studies adversarial attacks for financial systems that rely on ML systems.**

**Rating:** 4
**Confidence:** 4

**Review:**

The paper studies the impact of adversarial attacks on a ML based system for forecasting stock prices. The authors leverage Twitter data in order to enhance stock price prediction. Then, by determining the sensitivity of the model when perturbing the inputs. Then, small changes are applied to the inputs and output is observed. The authors experimented with the Tesla stock price.

This is interesting work and the authors do good work at explaining the different parts. Generally it flows well.

A first comment I have is about the focus of the paper as it studies both feasibility of forecasting as well as adversarial attacks. But, it does not go in depth in neither of these topics. For example, the first part for me is very interesting and having more forecasting approaches as well as adding more features with the additions of time-series would be very valuable.

Another comment I have is that there is a naive assumption. The fact that before sentiment classification we do not do any filtering. This does not make sense for an industrial system. Having a filtering step would greatly benefit the pipeline. The authors could have invested some space on such an approach.

As you conclude that an adversarial trader would need to know the details for the forecasting model in order to attack would you be able to suggest an approach that would not require this knowledge?

A major issue of the paper is that the authors just do predictions in a single stock price. I am not sure how much we can trust the conclusions based on just a single stock. The authors need to add more datasets. It is not enough.

What about having other type of textual features apart from the sentiment? Do you think there is more space to explore here?

---

> ### Author Response · Authors · 2020-11-24
> **Scope of our work**
>
> We thank the reviewer for their time and comments.
>
> We will answer each comment separately:
> Regarding the scope of our work:The goal of our work is to showcase the threat vector that may exist when using stock price forecasting machine learning models on public data via adversarial crafting. Given the page limit of ICLR, we had to limit the work of each section of the pipeline to ensure that every part of the pipeline can be discussed. We hope our work serves as a benchmark for much more work within adversarial machine learning in high-risk domains such as finance, hence we believe that the depth of each section was meaningful enough to create a non-trivial forecasting model.
>
> Regarding filtering of tweets, traditionally text is not preprocessed when training BERT; BERT tokenizer does it for you. Hence we think that simple language filtering was adequate in our work. However, we do realize that more complex filtering such as filtering tweets with little financial weight would be more indicative of an industry standard pipeline. As a preliminary work in financial adversarial machine learning, we decided to omit this added complexity. Lastly, we did filter out tweets with less than 50 likes and 10 retweets, but found very similar results in model performance and adversarial crafting performance.
>
> Regarding considering only a single stock, we agree with the reviewers opinion.We have since started collecting tweets from other companies, however Twitter updated their API at the end of September, breaking our scraping tool (and many others). Unfortunately, this severely limits the range of data we can collect (only a month of data) during the review stage. This is not enough data to train any of our models.
>
> Regarding the knowledge of an adversary, there are a couple hypothetical scenarios. If you know nothing about your victim's internal system but suspect they use twitter for sentiment analysis, you can collect the buy and sell decisions they make on the market along with the tweets at that time. Using the tweets as inputs and their decisions as labels, you can try different models on this dataset to try to best approximate their model. As more information is known to you (type of architecture, filtering process, hyperparameters), the task is easier. This is analog to black-box attacks based on transferability in adversarial example research on images.
>
> Lastly, we agree that other features can be of use for stock forecasting. Based on previous work mentioned in our work (Bollen et al, 2011), we believe textual sentiment is a good feature to detect financial information from tweets if such information exists. We experimented with the average length of tweets as a covariate, but found it uninformative for stock forecasting. Regarding additional features (for example, the author of the tweet, verified status, etc), we limited our work to features/covariates that can be manipulable by any member of the general public. For example, although the tweets from the CEO of the company in question are likely to be more meaningful then the tweet of a random individual, this is not manipulable by a random user (or at least, not without breaching the security of Twitter itself). This is part of the reason why we used tweets as a feature for stock forecasting (instead of other stocks, commodities etc) as they are more easily manipulable by anyone.

---

### Decision · Program_Chairs · 2021-01-07
**Final Decision**

**Decision:**

Reject

**Comment:**

One reviewer is positive, but that review is not of high quality. The other reviewers agree that this paper is interesting, but has too many limitations to be accepted by a highly competitive venue such as ICLR.